# A Systematic Review and Meta-Analysis of Continuous Subcutaneous Insulin Infusion vs. Multiple Daily Injections in Type-2 Diabetes

**DOI:** 10.3390/medicina59010141

**Published:** 2023-01-10

**Authors:** Vasiliki Chatziravdeli, George I. Lambrou, Athanasia Samartzi, Nikolaos Kotsalas, Eugenia Vlachou, John Komninos, Athanasios N. Tsartsalis

**Affiliations:** 1Department of Orthopedics, General Hospital “Ippokrateion”, Konstantinoupoleos 49, 54642 Thessaloniki, Greece; 2Choremeio Research Laboratory, First Department of Pediatrics, National and Kapodistrian University of Athens, Thivon & Levadeias 8, 11527 Athens, Greece; 3University Research Institute of Maternal and Child Health & Precision Medicine, National and Kapodistrian University of Athens, Thivon & Levadeias 8, 11527 Athens, Greece; 4Department of Endocrinology Diabetes and Metabolism, Naval Hospital of Athens, Dinokratous 70, 11521 Athens, Greece; 5Department of Nephrology, Naval Hospital of Athens, Dinokratous 70, 11521 Athens, Greece; 6Department of Nursing, School of Health Sciences, University of West Attica, Ag. Spydironos 28, 12243 Athens, Greece

**Keywords:** type 2 diabetes mellitus, insulin daily injection, continuous subcutaneous insulin infusion, glycated hemoglobin, fasting plasma glucose

## Abstract

Diabetes mellitus (DM) has a growing prevalence worldwide, even in developing countries. Many antidiabetic agents are used to improve glycemic control; however, in cases of an insufficient outcome, insulin is administered. Yet, the timing of proper insulin administration is still a subject of intense research. To date, there have been no recommendations or guidelines for the use of continuous subcutaneous insulin infusion (CSII) in Type 2 Diabetes Mellitus (T2DM). In the present study, we have performed a meta-analysis to evaluate the use of CSII in patients with T2DM. An extensive literature search was conducted through the electronic databases Pubmed, Clinicaltrials.gov, and Cochrane Central Register of Controlled Trials (CENTRAL) from October 2019–May 2022, for interventional studies related to T2DMI and CSII versus multiple daily injections (MDI). We included articles published in the English language only, yielding a total of thirteen studies. We found better outcomes in patients receiving CSII, in regard to glycated hemoglobin (HbA1c) and total insulin dose. In contrast, fasting plasma glucose and body weight did not show statistically significant differences between the two groups. Our analyses showed that CSII could be beneficial in patients with T2DM in order to achieve their glucose targets.

## 1. Introduction

Diabetes mellitus (DM) is defined as “A heterogeneous group of disorders characterized by hyperglycemia and glucose intolerance” (MeSH ID: D003920). DM has a growing prevalence worldwide. It is believed that in 2050, one-third of the adult population will exhibit DM, even in developing countries [1,2,3]. Current guidelines suggest the use of insulin as basal bolus or continuous subcutaneous insulin infusion (CSII) in Type 1 diabetes mellitus (T1DM) and for hospitalized patients [4]. On the other hand, treatment for Type 2 diabetes mellitus (T2DM) consists of antidiabetic agents and insulin as basal, basal plus, or basal bolus administration [5]. In particular, most recent guidelines include the use of SGLT2 inhibitors, which regulate Na^+^-Glucose transport in the kidney [6], and GLP1R agonists, which are incretin mimetics or GLP1 analogues [7]. Recent guidelines indicate the co-administration of GLP1R agonists with SGLT2 inhibitors in cases of treatment intensification [8]. The reason for this approach is the proven cardiovascular benefit [8]. Recent evidence has shown that the combination of insulin with GLP1R agonist, provided a similar reduction of HbA1c to a “basal-plus” or a “full basal-bolus” insulin regimen while inferring reduced hypoglycemia and body weight [8,9]. However, recent guidelines concerning the use of modern antidiabetic drugs, either as monotherapies or in combination, are restricted by the Glomerular Filtration Rate (GFR) [8]. Therefore, CSII could be proven useful in cases of contra-indication for the use of antidiabetic agents.

During recent decades, technology in diabetes has rapidly developed and improved, particularly in the areas of CSII and continuous glucose monitoring (CGM). This practice is well-established in patients with T1DM. It mimics the physiologic functions of the human pancreas, with beneficial effects in lowering glycated hemoglobin (HbA1c), reducing hypo- and hyperglycemia, and improving the quality of life [10,11]. The CSII combines the basal rhythm of insulin excretion with prandial bolus requirements and notifies the individual if their blood glucose levels are decreasing or increasing [12].

Antidiabetic drugs are an established first-line treatment. When glycemic control goals are not achieved, insulin therapy is considered; however, the appropriate time for the transition is still unclear due to a fear of side effects and the patient’s age. Recent advances in technology may divert this difficulty. To date, technology has been used only in T1DM. The pathophysiology of T2DM differs, in that the pancreas is still excreting insulin, but there is evidence that CSII could improve glycemic control as it does in T1DM [13].

Currently, there are no recommendations or guidelines for the use of CSII in T2DM since there is an existing debate regarding the efficacy of the method for insulin administration [14]. Therefore, we have used the available data from the literature to perform a meta-analysis in order to evaluate the use of CSII in patients with T2DM.

## 2. Materials and Methods

### 2.1. Literature Search

A systematic search of the literature was conducted in NCBI Pubmed, Clinicaltrials.gov, and Cochrane Central Register for Controlled Trials databases for relevant studies. The references from relevant reviews on the subject were also screened. We used keywords through the evaluation of Medical Subject Headings (MeSH), namely, “type 2 diabetes”, “continuous insulin infusion”, “insulin pump”, “multiple insulin injection”, and “daily injections”, which and limited our search criteria to include clinical trials and randomized controlled trials (RCTs) in humans were that was applicable. Only research published in English was considered. The search was concluded in May 2022. The detailed search strategy for each database is presented in Table 1.

The flowchart of the study selection is presented in Figure 1. The methods and results of this review were carried out in accordance with the principles of Preferred Reporting Items for Systematic Reviews and Meta-Analyses (PRISMA) (Appendix A) [15].

### 2.2. Selection Criteria

The inclusion criteria consisted of interventional studies (clinical trials and randomized controlled trials (RCTs)) that included adult patients with T2DM who received insulin therapy either with CSII, using an implantable device, or multiple daily insulin injections (Table 2). Eligible studies needed to include at least one outcome of interest. The outcomes evaluated were glycated HbA1c, fasting plasma glucose, body weight, and total daily insulin dose. Regarding large RCTs with multiple publications, studies reporting outcomes of interest that came from the same sample were included.

The exclusion criteria consisted of observational studies, case reports, case series, Phase I/II pharmacokinetic, and dose-determination studies, in vitro studies, animal studies, studies on a pediatric population, studies with only T1DM patients, studies with no full text available, or studies where the full text did not provide adequate data for extraction.

### 2.3. Study Selection

Two reviewers (VC and GIL) independently conducted the literature search according to the pre-specified criteria. Duplicate results were removed manually at the initial stage, and the remaining results were screened for eligibility by Title & Abstract. In the final stage, the full text of the remaining studies was assessed for inclusion. Studies approved by at least one of the reviewers was considered eligible. Whenever there was a dispute, a third author (ANT) resolved the issue.

### 2.4. Data Extraction

Data extraction was performed by VC and GIL and then approved by ANT. For all studies, we extracted the following data: The name of the first author, year of publication, type/name of the study, the population characteristics (body mass index, age, baseline HbA1c), number of participants, intervention device, insulin type, dosage, route and frequency of administration, treatment duration, and outcome measures.

### 2.5. Data Analysis

Data were imported into an Excel spreadsheet, Microsoft Office 365. Results were reported as means and standard deviation (SD) for continuous variables. For quantitative synthesis, a meta-analysis of studies was conducted by importing data into Review Manager 5.4 software [16] (accessed 22 October 2022). To account for measurement unit differences and between-study differences in the way variables were calculated, the standardized mean difference was used. Most studies reported the mean difference from baseline for each group for each variable, but some reported baseline and final values of means and SDs. To calculate the standard deviation of the mean difference (*σ_diff_*), the following formula from Cochrane Handbook for Systematic Reviews [17] was used:(1)$$\sigma_{diff}=\sqrt{\sigma_{E}^{2}+\sigma_{C}^{2}-(2\rho \sigma_{E}\sigma_{C})}$$
where *σ_E_* is the standard deviation baseline, *σ_C_* is the Standard deviation final, and *ρ* is the correlation coefficient set at 0.8. Sensitivity analysis was conducted by excluding large studies and calculating the pooled effect of studies with parallel and crossover design, separately. The *I*^2^ test was applied to test for heterogeneity between studies, as well as chi squared with the respective *p*-value. An *I*^2^ > 50% is interpreted as increased variability between the effect sizes. The random effects model was used in the meta-analysis. The presence of publication bias was assessed by producing funnel plots for the outcomes of interest. Results were considered significant at the *p* < 0.05 level.

### 2.6. Risk of Bias

To assess the risk of bias (methodological quality) of each study included in the review, we used the revised Cochrane risk-of-bias tool for randomized trials (RoB2) [18] for parallel study design and for crossover design. A fixed set of domains of bias (bias arising from the randomization process, bias from deviations to the intended interventions, bias from missing data, bias from measurement of the outcome, and bias from selection of the reported result) focusing on different aspects of trial design, conduct, and reporting were assessed. Two independent reviewers (VC and GIL) evaluated the included articles, and any discrepancies were resolved through discussion.

## 3. Results

### 3.1. Search Results

Our original search yielded 456 results. Forty full text studies were screened after duplicates and studies based on the Title and Abstract were removed. The final number of studies that were eligible for qualitative and quantitative synthesis after full text assessment was 13. A detailed diagram of the process with reasons for exclusion is illustrated in Figure 1.

### 3.2. Study Characteristics

The total number of included studies was 13. Ten were RCTs with a parallel study design [19,20,21,22,23,24,25,26,27,28] and two had a crossover design [29,30]. There was one study [31] that included subgroup analysis from a larger trial [26]. Details of the characteristics of the main studies are presented in Table 3.

### 3.3. Risk of Bias Assessment

The results from the risk of bias assessment of the included studies are presented in Figure 2 and Figure 3. There were some concerns arising from the randomization process because detailed information about how the randomization was performed was not provided in some studies, in addition to some concerns regarding the blinding process. There was no blinding of participants because of the nature of the intervention (pump implantation), and in most studies, no blinding of the personnel as well. Data from most of the randomized population were available for analysis. Lastly, in the crossover studies, there was low concern regarding the carryover effect since an adequate wash-out period was allowed.

### 3.4. Meta-Analysis Results

CSII using a pump device was proven more effective in reducing HbA1c from baseline as compared to multiple daily insulin injections with a pooled effect of −0.26 (−0.42, −0.10) (*p*-value = 0.02) (Figure 4). The same effect persisted when RCTs with a parallel design were analyzed separately, but the analysis of RCTs with a crossover design did not demonstrate a significant pooled effect in HbA1c reduction (Table 4). No significant difference between fasting plasma glucose (Figure 5) and body weight change (Figure 6) was evident between the two insulin infusion modalities. The daily insulin dose required to achieve target glucose levels was significantly lower in the intervention group (CSII) than the comparator [−0.58 (−0.76, −0.40)] (Figure 7).

The presence of publication bias for the outcomes of glycated HbA1c, body weight change, and daily insulin dose was examined by producing funnel plots for each outcome and is presented in Figure 8, Figure 9 and Figure 10. There does not appear to be significant publication bias, as depicted in the funnel plots.

## 4. Discussion

The purpose of our study was to evaluate the efficacy of CSII as compared to MDI in patients with T2DM. Our study demonstrated that CSII improved HbA1c levels with the exception of one study by Berthe et al. (2007) [29]. The reason for this discrepancy could be the fact that technology in insulin pumps was different formerly. More recent technology resembles the pancreatic function in a more efficient way [11,14,18,28,31,32]. A recent study reported on improved effectiveness of CSII as compared to MDI in T2DM patients [12], likely due to differences in study design, population used, methods, and insulin pump devices from previous clinical studies.

Regarding fasting plasma glucose levels, no significant differences were found between the two methods of insulin delivery. A possible explanation for this finding can be derived from the fact that for both methods, the desired effect is to equilibrate glucose levels between night and day. This equilibration is achieved with both approaches, yet CSII has the advantage of facilitating nocturnal hypoglycemia avoidance [33]. Therefore, it is believed that assessing HbA1c levels functions as a better marker. As the latter measures the average glucose level, including pre- and post-meal levels, it offers more concise information about the glucose daily fluctuations [34]. Therefore, it can be hypothesized that the measurement of HbA1c concentrations may be a more efficient criterion for evaluating insulin administration methods such as CSII and MDI. On the other hand, one study reported better outcomes for insulin pumps as compared to multiple injections with respect to fasting plasma glucose levels [35]. In this study, insulin delivery was performed through pre-mixed insulin analogues, which usually do not provide the appropriate dose. Finally, studies investigating plasma glucose levels are few, while only three were eligible for inclusion in the present meta-analysis.

Our meta-analysis showed no significant differences with respect to body weight. This was possibly due to the fact that most patients with T2DM have to follow specific dietary programs, including daily exercise and the use of antidiabetic agents, irrespectively of the method of administration used. Interestingly, this result was not expected because, in clinical practice, it has been observed that patients under MDI, tend to reduce their daily food intake, under the fear of non-predictable hyper- or hypoglycemia. On the other hand, patients under CSII tend to feel more secure, since the device is able to monitor glucose levels and thus assist in the regulation of hyper- or hypoglycemia. However, a previous study by Wainstein et al. (2004) [30] reported significant differences between the two methods (CSII vs. MDI) with respect to body weight. In this study, obese diabetic subjects were included, whereas the patient cohort under MDI was further administered metformin, an antidiabetic agent known to help reduce body weight.

The total insulin dose was reduced in patients with CSII. This finding is in agreement with results obtained in T1DM patients, whereas it was under investigation for T2DM patients. Noteworthily, all studies used in the present meta-analysis reported similar results, i.e., a significant reduction of total insulin dosage for patients under CSII as compared to patients under MDI. These results indicate that insulin pumps resemble pancreatic function in a more specific manner as compared to MDI, hence in a more patient-specific manner [36]. In the present meta-analysis, we have found that CSII could likely be beneficial not only for T1DM patients but also for T2DM patients. This becomes more evident in diabetic patients under MDI, who do not comply or are non-responsive to therapy [35].

In conclusion, insulin delivery in patients with T2DM with insulin pumps could be beneficial in achieving glycemic targets. Furthermore, insulin pumps might be cost effective in T2DM, as well as in T1DM [37,38]. Interestingly, in a recent meta-analysis, studying the efficacy of U-500 as MDI, it was reported that there was a significant reduction of 1.59% in HbA1c levels, a significant body weight gain of 4.38 kg, as well as a significant increase in total insulin daily dose (TDD) by 51.9 units [39]. In contrast, a nonsignificant weight gain and TDD were observed for the administration of U-500 as CSII [39]. Finally, the same report indicated that the use of U-500 regular insulin “both as MDI and via CSII was not reported to be associated with severe hypoglycemia but was associated with an increase in patient satisfaction as well as in cost savings”, which is in agreement with our clinical experience [39].

Previous studies and meta-analyses have also investigated the topic of MDI vs. CSII in selected populations. A recent report suggested that CSII was more effective as compared to MDI, and in particular, its effect was enhanced with a simultaneous DPP-4 inhibitor or GLP1 agonist administration [40]. An older meta-analysis attempted to investigate the efficacy of CSII in T1DM and T2DM patients, where it was found that patients using CSII pumps manifested a statistically significant reduction in glucose variability, as compared to those using MDI [41]. Finally, two older reports indicated that the use of CSII, as compared to MDI, manifested none [42] to small benefits [43] for patients with T2DM.

## 5. Conclusions

In the present study, CSII appeared to be favorable with respect to HbA1c, as it appeared that patients receiving CSII had lower HbA1c levels, whereas the daily insulin dose for obtaining optimal glucose levels was lower in patients under CSII as compared to MDI. On the other hand, fasting plasma glucose and body weight did not manifest any significant differences between the two study groups. CSII and MDI appear to be similarly effective, with CSII exceeding the factors of HbA1c and insulin administration levels. Thus, it is possible that CSII could prove useful in the treatment of T1 and T2DM. However, further larger randomized controlled trials with more patients and for larger periods of time are needed in order to reach more solid conclusions.

## Figures and Tables

**Figure 1 medicina-59-00141-f001:**
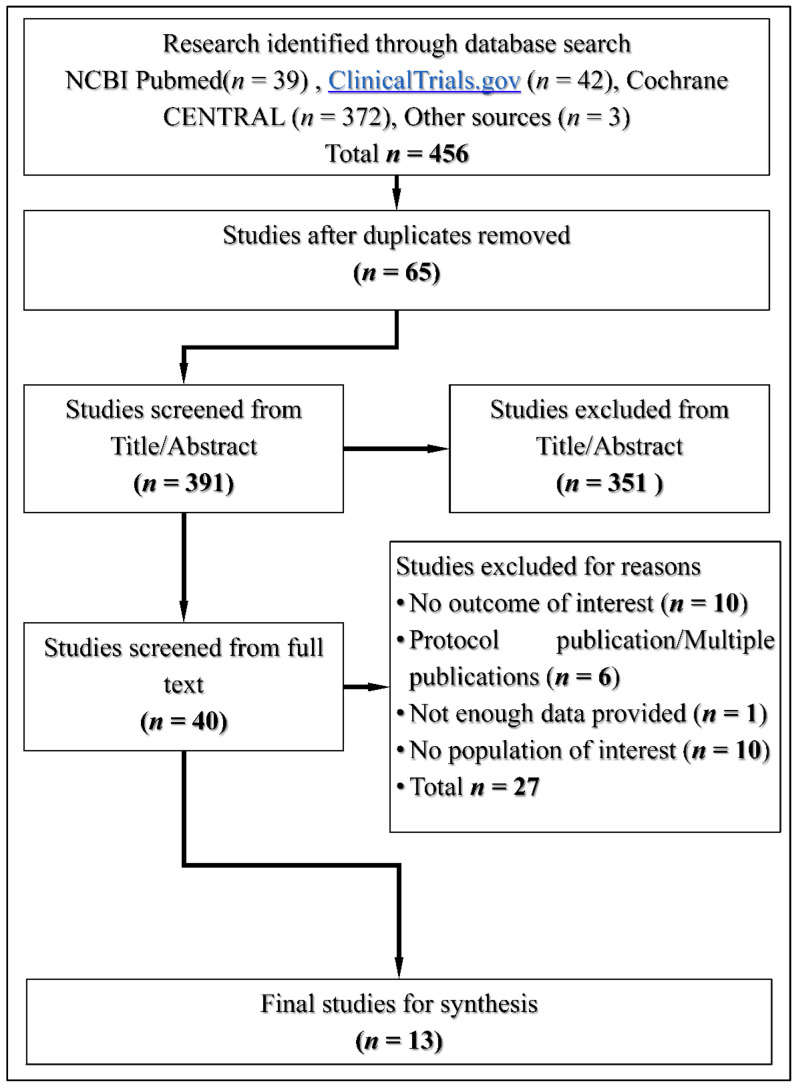
The flowchart of study selection.

**Figure 2 medicina-59-00141-f002:**
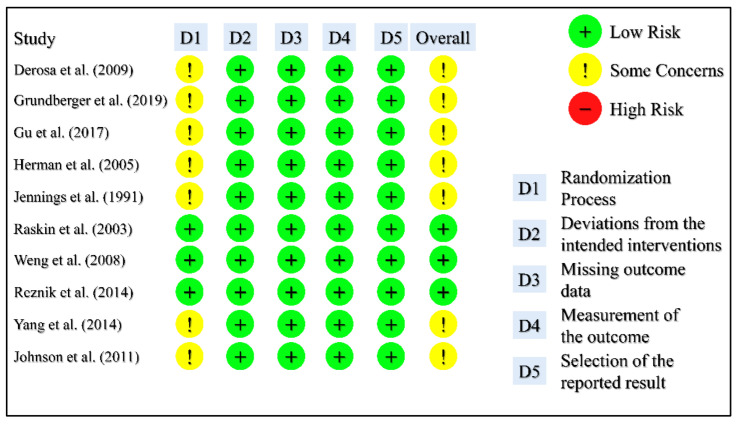
Risk of Bias Assessment Traffic Light Illustration for Studies with Parallel Design.

**Figure 3 medicina-59-00141-f003:**
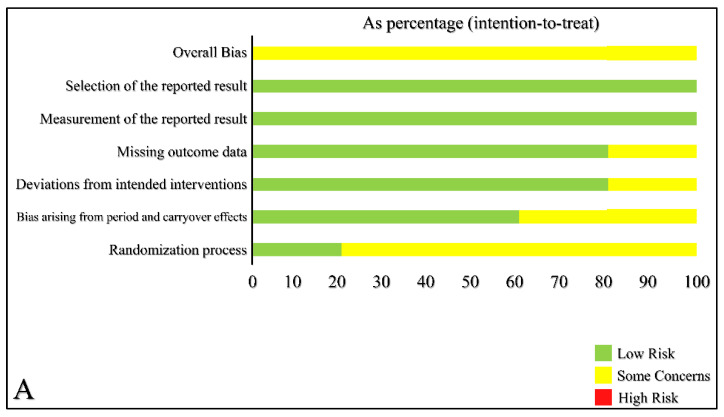
Risk of Bias Assessment Illustration as Percentage for Studies with Parallel Design (**A**), Risk of Bias Assessment Traffic Light Illustration for Studies with Crossover Design (**B**).

**Figure 4 medicina-59-00141-f004:**
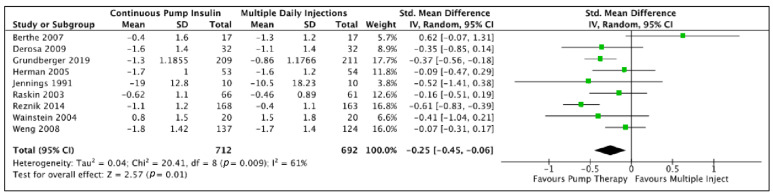
Meta-analysis results and Forest plot for Glycated HbA1c.

**Figure 5 medicina-59-00141-f005:**
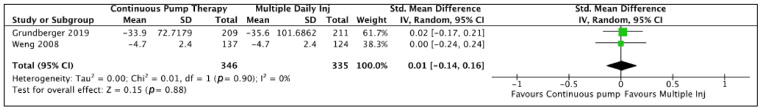
Meta-analysis results and Forest plot for Fasting Plasma Glucose.

**Figure 6 medicina-59-00141-f006:**
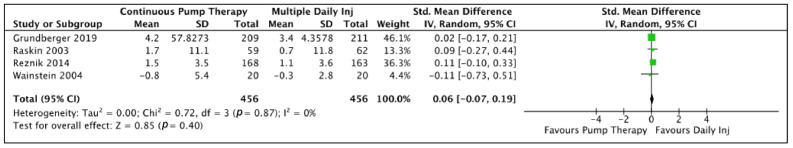
Meta-analysis results and Forest plot for body weight.

**Figure 7 medicina-59-00141-f007:**
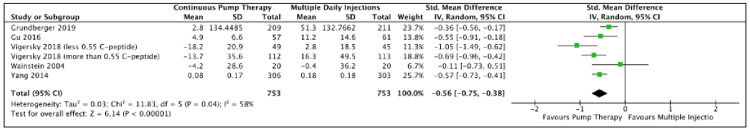
Meta-analysis results and Forest plot for Daily Insulin Dose.

**Figure 8 medicina-59-00141-f008:**
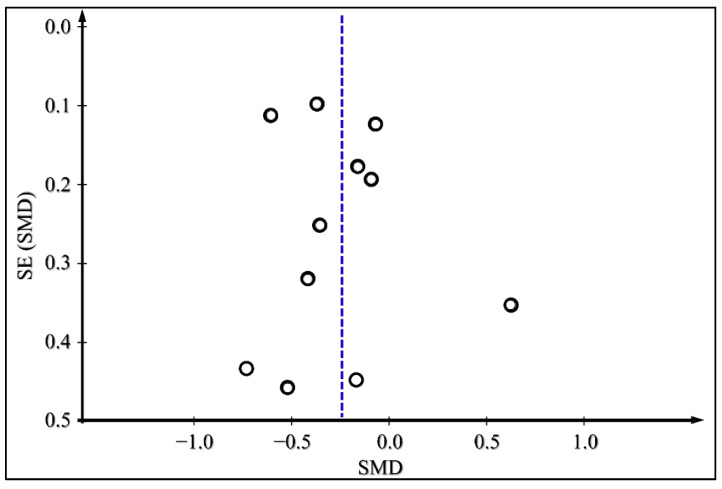
Funnel plot for Publication Bias Assessment for Glycated HbA1c.

**Figure 9 medicina-59-00141-f009:**
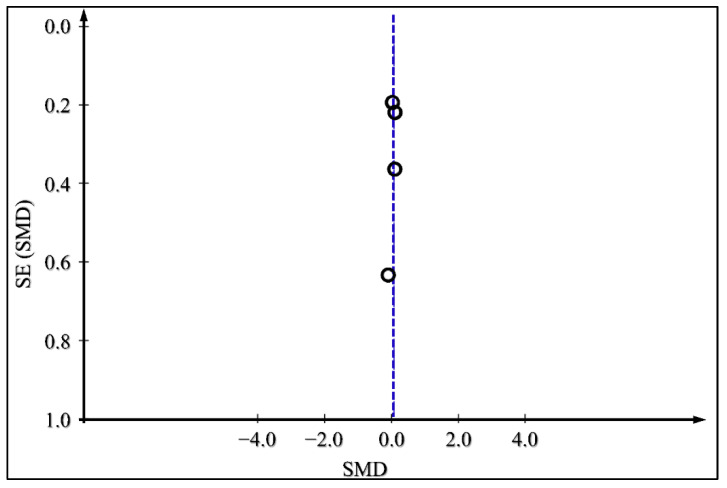
Funnel plot for Publication Bias Assessment for body weight.

**Figure 10 medicina-59-00141-f010:**
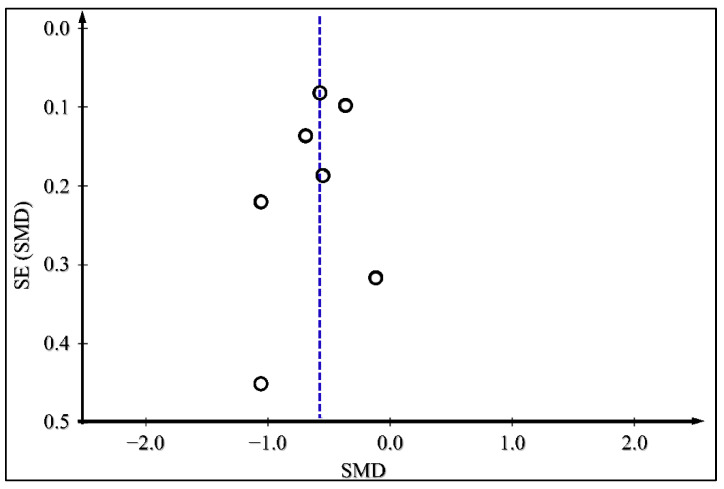
Funnel plot for Publication Bias Assessment for Daily Insulin Dose.

**Table 1 medicina-59-00141-t001:** Search strategy with the respective keywords used for the present analysis (numbers in parenthesis indicate the number of publications found).

Database	Search Strategy
NCBI Pubmed (*n* = 39)	((((type 2 diabetes[Title/Abstract]) AND (insulin pump[Title/Abstract])) OR (continuous subcutaneous infusion[Title/Abstract])) AND (daily injections[Title/Abstract])) OR (multiple insulin injection[Title/Abstract]) Filters Clinical trial and Randomized controlled trial
ClinicalTrials.gov (*n* = 42)	Type 2 Diabetes AND insulin Pump Filters Completed studies
Cochrane CENTRAL (*n* = 372)	type 2 diabetes in Title Abstract Keyword AND continuous insulin in Title Abstract Keyword AND multiple injection in Title Abstract Keyword—(Word variations have been searched)

**Table 2 medicina-59-00141-t002:** Inclusion criteria of the present study.

Population	Intervention	Comparator	Outcome
T2DM	Continuous, Subcutaneous Insulin Infusion	Multiple Daily Insulin Injection	Difference in HbA1c Difference in Fasting plasma glucose Difference in Body Weight Difference in Total daily insulin dose

**Table 3 medicina-59-00141-t003:** Table of Characteristics of Included Studies.

Author	Study Design	Population			Intervention/ Comparator	Pump Type	Insulin type	No Intervention	No Comparator	Route, Dose, Frequency I/C	Treatment Duration I/C MD (IQR)	Outcomes
Berthe et al. (2007) [29]	Open label RCT crossover	33.7 (4.6)	9 (1.6)	55 (6)	CSII/MDI	Medronic 508	Lispro plus NPH	17	17	70% daily + 30% prandial bolus/ 3 daily inj 50/50	12 w	HbA1c, capillary blood glu, hyperglycemic AUC, pt satisfaction, chol, Tg
Derosa et al. (2009) [19]	Randomized case-control trial (Type 1 & 2)	29.5(5.1)/29.8(5.4)	9.2(2)/9.3(2.1)	49.8(14.6)/50.4(14.2)	CSII/ MDI		Lispro /glargine	32	32	47 UI 50–50/33 UI 3 shots daily lispro + 22 UI 1 shot glargine ins	12 m	HbA1c, fasting plasma glu, post-prandial glu, total chol, HDL, Tg
Grunberger et al. (2019) [20]	Open label RCT-parallel (VIVID study)	39.3(5.6)/40.1(5.8)	8.75(1.03)/ 8.77(1.08)	57.6(10.3)/56.7(10.1)	CSII/MDI	Omnipod DASH U-500	U-100 rapid/U-500 R + other glu lowering agents	209	211	50–50/3 daily inj	26 w	HbA1c, fasting plasma glu, proportion achieving target HbA1c
Gu et al. (2016) [21]	Open label RCT parallel NCT01921322	25(3.1)/25(3.3)	10(1.6)/10(1.2)	51(10.2)/49(9.6)	CSII/MDI	Medronic MiniMed Paradigm sensor augmented pump	Novo Nordisk A/S fast and long acting	57	61	50–50 fast acting/3 daily inj + 1 bed-time inj		Time to achieve blood glu levels, pts achieving target glu, AUC,
Herman et al. (2005) [22]	RCT parallel	32.5(5.8)/31.8(5.8)	8.4(1.1)/ 8.1(1.2)	66.6(5.9)/66.2(4.5)	CSII/MDI	Medronic MiniMed 508	Lispro and glargine	53	54	50–50/3 daily +1 before bed-time	12 m	HbA1c, QoL
Jennings et al. (1991) [23]	RCT parallel		64.5/62.5	58/61	CSII/MDI		Regular and NPH	10	10	2 daily	4 m	HbA1c, Fasting glu, capillary blood glu, chol, Tg, HDL, satisfaction
Johnson et al. (2011) [24]	RCT Parallel	33.5(5.7)/31.8(5.9)	8.3(1.1)/8.1(1.3)	66(6)/66(4.6)	CSII/MDI	Medronic MiniMed 508	Lispro /glargine	53	54		12 m	Mean day glu, mean pre-prandial glu, AUC high, AUC-low
Reznik et al. (2014) [26]	Open label-RCT parallel with single arm crossover OpT2mise NCT01182493	33.5(7.5)/33.2(7)	9	55.5(9.7)/56.4(9.5)	CSII/MDI	Medronic MiniMed Paradigm Veo	Lispro or aspart or glulisine & glargine or detemir	168	163	50–50/Inj at investigator’s clinical practice	6 m/6 m crossover of MDI to CSII	HbA1c, AUC hypoglycemia/hyperglycemia
Vigersky et al. (2018) [31]	OpT2mise- subgroup analysis				CSII/MDI according to C-peptide level & Age							HbA1c, TDD, satisfaction
Raskin et al. (2003) [25]	Open label RCT parallel	32.2(4.2)/32.2(5.1)	8.2(1.4)/ 8(1.1)	55.1(10.2)/56(8.18)	CSII/MDI	Medronic MiniMed 507C	Insulin aspart & NPH	66	61	Ins aspart continuous/ Ins aspart after meals + once or twice long acting ins	24 w	HbA1c, BG, TDD, satisfaction
Wainstein et al. (2004) [30]	RCT crossover	30–45	>8.5%	30–70	CSII/MDI	Medronic MiniMed	Lispro /Regular ins or Humulin R & NPH	20	20	4 daily inj	18 w/18 w	HbA1c, AUC, chol, HDL, LDL, Tg, C-pept, weight
Weng et al. (2008) [27]	RCT parallel NTC00147836	25.1(3)/ 24.4(2.7)/25.1(3.3)	9.8(2.3)/9.7(2.3)/9.5(2.5)	50(11)/51(10)/52(9)	CSII/ MDI/ oral agents		Human ins (Novo Nordisk)/ Novolin-R & NPH/ gliclazide or metformin + gliclazide	137	124/121	50–50/ 30–20-20–30/ 80 mg twice daily gliclazide ± 0.5–2 g metformin	12 m	Fasting plasma glu, β-cell function, HbA1c, Tg, chol, LDL, HDL,
Yang et al. (2014) [28]	RCT parallel	24.41(3.63)/24.89(3.48)	10.46(2.12)/10.34(2.15)	51.38(11.74)/50.58(12.68)	CSII/MDI	Medronic	Ins aspart/ Human ins short-acting (Novo Nordisk) & NPH	306	303	40–60/ 3 times/d fast + 2 times/d long (40–60)	12 w	Days to achieve target glu, BG levels, TDD, hypoglycemia

**Table 4 medicina-59-00141-t004:** Pooled results of meta-analysis and sensitivity analysis for each outcome.

				*Test of Association*				*Test of Heterogeneity*		
*Interventions*	Outcomes	Subgroups	Effect Sizes	Pooled SMD(CI)	*p*-Value	Model	Z-Test	X^2^	*p*-Value	I^2^(%)
*Continuous Subcutaneous Insulin Infusion v.s Multiple Daily Injections*	Difference in HbA1c		12	−0.26(−0.42, −0.10)	0.002	RE	3.11	22.14	0.02	50
		Sensitivity Analysis								
		After 2 larger studies removed	10	−0.14(−0.28,0)	0.05	RE	1.99	9.05	0.43	1
		Parallel design only	7	−0.28(−0.47, −0.10)	0.002	RE	3.08	14.76	0.02	59
		Crossover design only	5	−0.17(−0.57, 0.23)	0.41	RE	0.82	6.73	0.15	41
	Difference in Body Weight		6	0.20(−0.16, 0.55)	0.28	RE	1.09	23.72	0.0002	79
		Parallel design only	4	0.06(−0.07, 0.19)	0.34	RE	0.95	0.42	0.94	0
	Fasting Plasma Glucose Difference		3	−0.01(−0.14, 0.13)	0.94	RE	0.08	0.36	0.83	0
	Daily Insulin Dose Difference		7	−0.58(−0.76, −0.40)	<0.00001	RE	6.41	13.12	0.04	54
		Parallel design only	5	−0.54(−0.69, -0.40)	<0.00001	RE	7.38	5.93	0.2	32

SMD: Standardized mean difference; CI: Confidence interval; RE: Random effect; HbA1c: Glycated hemoglobin; vs.: Versus; *p*-value< 0.05 is considered significant; I^2^ >75% is considered significant heterogeneity.

## Data Availability

Not applicable.

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
