# Peer review of "A Systematic Review and Meta-Analysis of Continuous Subcutaneous Insulin Infusion vs. Multiple Daily Injections in Type-2 Diabetes"

_medicina, 2023, doi:10.3390/medicina59010141_

Round 1

Reviewer 1 Report

The study by Chatziravdeli et al. addresses an important topic. Despite being well-presented, I have some comments that need to be addressed.

1.      Many abbreviations have been introduced into the abstract and the manuscript itself, especially for the names of types of insulin therapy, and are not used anyway. Please use abbreviations.

2.      Words such as "suffer from" should not be used in relation to patients. Please replace with a synonym.

3.      Please use paraphrases for sentences written in the abstract, as most sentences in are identical to those in the manuscript.

4.      Before the topic of the paper was considered, was the database of systematic reviews, e.g. prospero, searched so as not to duplicate a topic on which other Researchers are working?

5.      Please specify that you mean e.g. body weight, diabetes mellitus etc.

6.      Links pasted into the text should be considered as references to the bibliography.  (see 2.5. Data Analysis)

Author Response

Review #1 (Round 1)

Open Review

(x) I would not like to sign my review report

( ) I would like to sign my review report

English language and style

( ) English very difficult to understand/incomprehensible

( ) Extensive editing of English language and style required

( ) Moderate English changes required

(x) English language and style are fine/minor spell check required

( ) I don't feel qualified to judge about the English language and style

Yes

Can be improved

Must be improved

Not applicable

Does the introduction provide sufficient background and include all relevant references?

( )

(x)

( )

( )

Are all the cited references relevant to the research?

( )

(x)

( )

( )

Is the research design appropriate?

(x)

( )

( )

( )

Are the methods adequately described?

(x)

( )

( )

( )

Are the results clearly presented?

( )

(x)

( )

( )

Are the conclusions supported by the results?

(x)

( )

( )

( )

Comments and Suggestions for Authors

The study by Chatziravdeli et al. addresses an important topic. Despite being well-presented, I have some comments that need to be addressed.

  1. Many abbreviations have been introduced into the abstract and the manuscript itself, especially for the names of types of insulin therapy, and are not used anyway. Please use abbreviations.

Response: We thank the reviewer for the thorough review. We have replaced terms with abbreviations throughout the text.

  1. Words such as “suffer from” should not be used in relation to patients. Please replace with a synonym.

Response: Thank you for your comment. We have replaced the “suffer from” with the synonym, “…they will exhibit…” throughout the text.

  1. Please use paraphrases for sentences written in the abstract, as most sentences in are identical to those in the manuscript.

Response: We have modified the abstract as suggested (please refer to the “Abstract” section for modifications).

  1. Before the topic of the paper was considered, was the database of systematic reviews, e.g. Prospero, searched so as not to duplicate a topic on which other Researchers are working?

Response: We thank the reviewer for the insightful comment. We have, indeed, searched the Prospero database and at the time of conducting our metanalysis, there were no similar works. In particular, at the time we conducted our research there were no similar works. In addition, during ’22 an additional 20 works referred to the topic, yet with no similarities to our work.

  1. Please specify that you mean e.g. body weight, diabetes mellitus etc.

Response: We have added the definition for diabetes in the first sentence of the “Introduction” section. Further on, we have used the BMI, instead of body weight for our metanalysis.

  1. Links pasted into the text should be considered as references to the bibliography (see 2.5. Data Analysis).

Response: Links were moved to the “References” section as suggested.

Reviewer 2 Report

The authors conducted a meta-analysis to evaluate the usefulness of CSII in patients with type 2 diabetes. The authors have shown that patients with type 2 diabetes treated with CSII had lower HbA1c and lower total daily dose of insulin than patients treated with MDI. However, there have already been several reports examining whether CSII or MDI is more useful for patients with type 2 diabetes. It is also well known that total daily dose of insulin is lower in CSII than in MDI. Furthermore, advances in antihyperglycemic agents such as GLP-1 receptor agonists and dual GIP and GLP-1 receptor agonists are expected to reduce the proportion of type 2 diabetes treated with MDI or CSII. In addition, I have some comments on this study

1. Studies using insulin preparations that are no longer used in routine clinical practice, such as lente insulins, should be excluded from this study. In addition, premixed insulins and lente insulins should not treated as MDI.

2. Similarly, U-500R should not be included in this study.

3. Further discussion is needed, citing the results of other meta-analyses of patients with type 2 diabetes.

Author Response

Review #2 (Round 1)

Open Review

( ) I would not like to sign my review report

(x) I would like to sign my review report

English language and style

( ) English very difficult to understand/incomprehensible

( ) Extensive editing of English language and style required

( ) Moderate English changes required

( ) English language and style are fine/minor spell check required

(x) I don't feel qualified to judge about the English language and style

Yes

Can be improved

Must be improved

Not applicable

Does the introduction provide sufficient background and include all relevant references?

(x)

( )

( )

( )

Are all the cited references relevant to the research?

( )

( )

(x)

( )

Is the research design appropriate?

(x)

( )

( )

( )

Are the methods adequately described?

(x)

( )

( )

( )

Are the results clearly presented?

(x)

( )

( )

( )

Are the conclusions supported by the results?

(x)

( )

( )

( )

Comments and Suggestions for Authors

The authors conducted a meta-analysis to evaluate the usefulness of CSII in patients with type 2 diabetes. The authors have shown that patients with type 2 diabetes treated with CSII had lower HbA1c and lower total daily dose of insulin than patients treated with MDI. However, there have already been several reports examining whether CSII or MDI is more useful for patients with type 2 diabetes. It is also well known that total daily dose of insulin is lower in CSII than in MDI. Furthermore, advances in antihyperglycemic agents such as GLP-1 receptor agonists and dual GIP and GLP-1 receptor agonists are expected to reduce the proportion of type 2 diabetes treated with MDI or CSII. In addition, I have some comments on this study.

  1. Studies using insulin preparations that are no longer used in routine clinical practice, such as lente insulins, should be excluded from this study. In addition, premixed insulins and lente insulins should not treated as MDI.

Response: We thank the reviewer for this insightful comment. We agree with the reviewer that this type of insulin is not commonly applied in the clinical practice, yet we still administer the aforementioned NPH insulin in pregnant patients. Further on, there was only one study concerning the administration of lente insulin, which did not affect our results, as it manifested comparable efficacy as compared to MDI. Finally, NPH, as well as lente insulins, have been in use for long periods which obliged us to include all possible studies for a more complete understanding of the phenomenon. Interestingly, all approaches agreed on the outcome, namely the favorable use of CSII.

  1. Similarly, U-500R should not be included in this study.

Response: This type of therapy was introduced for reducing the total insulin dose, by administrating a more condensed solution which alleviated a series of problems, including insulin metabolism, reduction of the amount of liquid thus reducing the total number of administration, reduction of total injections applied. Therefore, we thought that these studies should be included in the present analysis, since they are widely used and a comparison between them and CSII was necessary. Finally, it became evident that despite all differences in type of administration, concentration or injection numbers the results were not affected and still the CSII treatment appeared more favorable. We have added a small paragraph in the “Discussion” section concerning previous metanalyses for U-500 (please refer to “Discussion” page 12).

  1. Further discussion is needed, citing the results of other meta-analyses of patients with type 2 diabetes.

Response: We thank the reviewer for the suggestion. We have added a paragraph in the “Discussion” section as suggested (please refer to page 11).

Round 2

Reviewer 2 Report

Unfortunately, the authors failed to understand the intent of the reviewers and modify their paper. As the authors describe, intensive insulin therapy is used when oral medications are contraindicated. In addition, intensive insulin therapy is also used in type 2 diabetes when endogenous insulin secretion is depleted or when diabetes medications do not provide adequate glycemic control. Therefore, as you know, multiple daily insulin injections and continuous subcutaneous insulin injections are also important treatments for type 2 diabetes. As the authors describe, NPH may still be used in daily clinical practice. However, lente insulin is not even manufactured today. Furthermore, MDI is a common treatment regimen for diabetes, typically consisting of three or more injections of insulin per day. Therefore, today insulin therapy with pre-mixed insulin or lente insulin is usually treated as a non-MDI. 

Type 2 diabetes should be listed as "type 2", not "type II".

Author Response

Open Review

( ) I would not like to sign my review report

(x) I would like to sign my review report

English language and style

( ) English very difficult to understand/incomprehensible

(x) Extensive editing of English language and style required

( ) Moderate English changes required

( ) English language and style are fine/minor spell check required

( ) I don't feel qualified to judge about the English language and style

Yes

Can be improved

Must be improved

Not applicable

Does the introduction provide sufficient background and include all relevant references?

( )

( )

(x)

( )

Are all the cited references relevant to the research?

( )

( )

(x)

( )

Is the research design appropriate?

( )

( )

(x)

( )

Are the methods adequately described?

(x)

( )

( )

( )

Are the results clearly presented?

(x)

( )

( )

( )

Are the conclusions supported by the results?

( )

( )

(x)

( )

Comments and Suggestions for Authors

Unfortunately, the authors failed to understand the intent of the reviewers and modify their paper. As the authors describe, intensive insulin therapy is used when oral medications are contraindicated. In addition, intensive insulin therapy is also used in type 2 diabetes when endogenous insulin secretion is depleted or when diabetes medications do not provide adequate glycemic control. Therefore, as you know, multiple daily insulin injections and continuous subcutaneous insulin injections are also important treatments for type 2 diabetes. As the authors describe, NPH may still be used in daily clinical practice. However, lente insulin is not even manufactured today. Furthermore, MDI is a common treatment regimen for diabetes, typically consisting of three or more injections of insulin per day. Therefore, today insulin therapy with pre-mixed insulin or lente insulin is usually treated as a non-MDI.

Response: Following the reviewer’s suggestions we have made the following changes in our manuscript: a) we have modified the flow chart accordingly to exclude the studies that did not meet the criteria. In particular, the studies removed were Blackshear et al. (1989), Chulp et al. (2018) and Saudek et al. (1996), b) studies were also removed from Table 3, c) Figures 2 and 3 were modified accordingly, d) Forrest plots were also corrected, where the aforementioned studies were removed from Figures 4, 5, 6, 7 and e) Funnel plots were also modified accordingly in Figures 8, 9, 10.

Finally, the manuscript was extensively checked and English language was proof-read.

All new changes can be found in the file “16.Chatziravdeli et al 2022_DM_Pumps_Metanalysis_R2_with_TrackChanges.docx”, while the file “16.Chatziravdeli et al 2022_DM_Pumps_Metanalysis_R2_without_TrackChanges.docx”, contains a clear-view of the final manuscript.

Type 2 diabetes should be listed as "type 2", not "type II".

Response: We were not able to find the nomenclature “Type II” in our manuscript, yet this nomenclature was mentioned in the “References” section and in one of our tables, which was directly reproduced as it appeared in the PubMed database. We have corrected the nomenclature in order to appear as the reviewer required.

Round 3

Reviewer 2 Report

The authors have revised the paper appropriately, and I have no further comments on this manuscript.